# Impact of point-of-care tests in community pharmacies: a systematic review and meta-analysis

Ali Albasri ![ORCID],[1] Ann Van den Bruel,[2,3] Gail Hayward ![ORCID],[1] Richard J McManus ![ORCID],[1] James Peter Sheppard ![ORCID],[1] Jan Yvan Jos Verbakel ![ORCID] [1,2,3]

¹Nuffield Department of Primary Care Health Sciences, University of Oxford, Oxford, UK
²Academic Centre for General Practice, Department of Public Health and Primary Care, KU Leuven, Leuven, Flanders, Belgium
³EPI-Centre, Department of Public Health and Primary Care, KU Leuven, Leuven, Flanders, Belgium

**Correspondence to**
Professor Jan Yvan Jos Verbakel;
jan.verbakel@phc.ox.ac.uk

## ABSTRACT

**Objectives** To summarise the literature regarding the use of point-of-caretest (POCT) in pharmacies versus control/usual care.

**Design and setting** Systematic review and random-effects meta-analysis in community pharmacy.

**Data sources** MEDLINE, Cochrane Central Register of Controlled Trials, Embase, ClinicalTrial.gov and Web of Science databases were searched.

**Eligibility criteria** Articles were included if they: involved a POCT conducted by a community pharmacist, member of pharmacy staff or local equivalent; measured a clinically relevant outcome for example, clinical parameter monitoring. No clinical condition or language limits were set.

**Patient and public involvement** No patient involvement.

**Data extraction and synthesis** Data were independently extracted by two members of the review team to capture changes in clinical care that resulted from the use of the POCTs. The methodological quality of included studies was assessed, using the Cochrane Risk of Bias tool and Newcastle-Ottawa scale.

**Results** Thirteen of the 1584 articles found were included in the meta-analyses. Studies covered four therapeutic areas: targeted anti-malarial therapy (n=3 studies), glycatedhaemoglobin (HbA1c) in diabetes (n=2 studies), lipid control (n=3 studies) and international normalised ratio (INR) control in patients taking warfarin (n=5 studies). POCT in pharmacies reduced the risk of receiving antimalarial treatment when not clinically indicated (riskratio 0.34, 95% CI 0.31 to 0.37). Lipid and HbA1c control appeared largely unaffected by pharmacy POCTs, and the impact on INR time-in-therapeutic-range was inconclusive.

**Conclusions** Only 4 out of 13 included studies used a gold-standard randomised controlled trial (RCT) design, limiting our ability to conclusively determine the clinical utility of POCT conducted in pharmacies. Further RCTs are needed, particularly in areas such as upper respiratory tract infections, which have gathered momentum among service commissioners in recent years.

**PROSPERO registration number** CRD42017048578.

## Strengths and limitations of this study

► This review provides a timely and comprehensive overview of the current evidence related to point-of-caretest (POCT) in pharmacies.
► The majority of included studies were observational and were generally of poor methodological quality.
► Pooling of data from a small number of studies per comparison led to high levels of observed statistical heterogeneity across a majority of comparisons.
► The review places into context the need for evidence-based policymaking regarding the use of POCT.

made available within the same clinical visit to support clinical decision-making.[1] These tests have the potential to save clinical time and improve patient access to care in the form of diagnoses, medications or dose amendments.[2]

Interest in the use of POCTs in different healthcare settings is increasing and is expected to grow significantly in the years to come.[3] In 2016, the 'Community Pharmacy Forward View' was published as a response by the pharmacy sector to the then 'NHS FiveYear Forward View', and suggested that diagnostics and POCTs should be made routinely available in pharmacy settings.[4]

Given the current strain on primary healthcare services,[5] the provision of POCTs has become more commonplace in UK community pharmacies, with particular emphasis on the potential for POCTs to aid both acute condition diagnosis and long-term condition management.[6] In 2016, National Health Service (NHS) England approved a 'test and treat' service at a large pharmacy chain for patients presenting with sore throats, in an attempt to curb inappropriate antibiotic prescriptions and reduce burden on general practice.[7] However, the evidence behind the use of POCTs in pharmacies appears to be from either pilot studies, non-randomised studies or studies with no comparator

## INTRODUCTION

A point-of-care test (POCT) can be defined as a test performed by a qualified member of staff nearby the patient, where results are

groups.[8] The evidence-base for implementing POCTs remains a concern more generally given that studies tend to focus on test performance (method comparison with central laboratory testing) rather than clinical or healthcare utilisation outcomes.[9]

While previous work has focussed on the analytical quality of POCTs used by community pharmacists,[10] this paper presents the findings of a systematic review and meta-analysis assessing the clinical impact of POCTs in community pharmacies on clinical outcomes and healthcare processes.

## METHODS
### Search strategy
A comprehensive search strategy in MEDLINE, Cochrane Central Register of Controlled Trials, Embase, ClinicalTrial. gov and Web of Science was devised. An example of the MEDLINE search terms can be found in online supplementary table 1. Relevant articles from inception to 24 April 2019 were searched in addition to references of relevant reviews and articles that met our selection criteria. No language limits or study design filters were applied.

### Selection of studies and inclusion criteria
Two members of the review team (AA and JYJV) independently reviewed titles, abstracts and full texts. Studies screened by title, abstract and full-text were eligible for inclusion if they met all of the following criteria:
1. A POCT conducted by a community pharmacist or member of community pharmacy staff (ie, pharmacy technician, healthcare assistant or local equivalent).
2. Clinically relevant outcome measures reported, for example, change in clinical care such as: referral, admission to hospital, morbidity, mortality or rate of diagnosis, time in therapeutic range, duration of illness.
3. Patients of all ages presenting to a community pharmacy for any medical condition.

Randomised controlled trials, non-randomised but experimental and controlled studies including before-and-after and retrospective cohort studies were included in this review. Systematic reviews were excluded but their reference lists were searched for relevant primary studies.

Studies were excluded if any of the following criteria applied:
1. Were diagnostic accuracy studies (focussing only on the performance of one or more point-of-care tests vs a central lab test).
2. Included only hospital inpatients.
3. Studies without a control group or comparator.
4. Patients self-testing or tests that were taken away by patients (to test at home, for example).
5. Included a POCT as part of a wider intervention, such that the effect of the POCT alone could not be ascertained.

### Outcomes measured
The primary outcome of this review was the impact of POCT on clinically relevant outcomes such as changes to treatment, disease marker monitoring, referrals, admissions to hospital, morbidity, mortality, time to diagnosis, time in therapeutic range or duration of illness.

### Data extraction
Data were independently extracted and verified by two members of the review team (AA and JYJV). Data were extracted to capture changes in clinical care that resulted from the use of the POCTs. The following data were extracted from the primary studies where available: referral or admission to other healthcare providers, mortality, morbidity, time in therapeutic range, percentage of patients reaching therapeutic targets such as cholesterol and glycatedhaemoglobin (HbA1c), resulting medication recommendations or appropriateness of medication recommendations.

### Quality assessment
The methodological quality of included studies was assessed independently by two authors (AA and JYJV). Randomised trials were assessed using the Cochrane Risk of Bias tool[11] and included analysis of randomisation, allocation concealment, comparison of baseline characteristics and blinding. For non-randomised but experimental and controlled studies, the Cochrane Risk of Bias tool for observational studies was used. Case-control studies were assessed using the Newcastle-Ottawa scale.[12]

### Data synthesis
Meta-analyses were conducted separately for randomised controlled trials and non-randomised studies whenever three primary studies or more were available per prespecified analysis. Data were analysed using a random-effects model due to expected heterogeneity in study designs and populations.[13] Analyses were grouped according to the condition to which the POCT related. Data were combined using the Review Manager (RevMan) V.5 software. For outcomes where meta-analysis was not possible, results were described qualitatively.

Where statistical heterogeneity was detected, possible contributing factors such as the setting or operator, the patient population and/or other methodological characteristics were investigated in sensitivity analyses where possible. Where data allowed, publication bias was assessed via Egger's test to check for small study effects.[14]

Data are presented as a proportion of each study population, means with SD or 95% CIs unless otherwise stated.

The study protocol was registered on PROSPERO: International Prospective Register of Systematic Reviews and can be found online (http://www.crd.york.ac.uk/prospero).

### Patient and public involvement
No patient involvement.

## RESULTS
### Study selection
After removal of 619 duplicate records, 1584 studies were identified from the literature searches and one

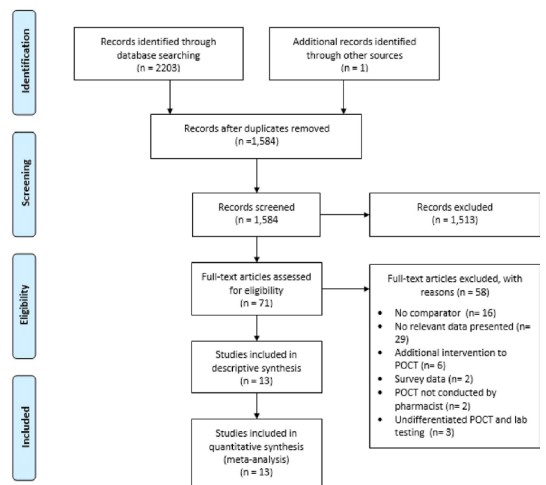

**Figure 1** Preferred Reporting Items for Systematic Reviews and Meta-Analyses flow diagram.

additional record was found from citation searches. After title and abstract screening 1513 studies were excluded leaving 71 studies to be screened by full-text. Of these, 58 studies were excluded for the reasons stated in the Preferred Reporting Items for Systematic Reviews and Meta-Analyses flow diagram (figure 1) leaving 13 studies eligible for inclusion.

## Study characteristics

Table 1 shows the characteristics of included studies: seven were observational (pre/post design),[15–21] four were randomised controlled trials (RCTs),[22–25] one a prospective controlled staggered parallel design study and one a retrospective cohort study.[26] Studies were conducted in the USA (n=5), Canada (n=2), Australia (n=1), New Zealand (n=1), Ghana (n=1), Nigeria (n=1), India (n=1), Uganda (n=1) and included data from a total of 23 149 patients. All POCTs were conducted by a community pharmacist(s) or local equivalent that received training in both delivering the POCT and in the subsequent treatment recommendation.

None of the RCTs or observational studies charged the patient directly for the POCT. Patients were most commonly recruited into the observational studies via clinician referral or through pharmacy list searches, with only one of the seven observational studies recruiting patients opportunistically. Three of the RCTs recruited patients opportunistically on presentation to the pharmacy, with the fourth recruiting eligible patients by invitation from a clinical list.

## Quality assessment

The overall methodological quality (online supplementary figures 1-3) was moderate across the five prospective controlled trials, with three studies exhibiting a high risk of detection bias (lack of blinding of the outcome assessors) and an unclear risk of reporting bias (no study protocol available).[22 24 27] The non-randomised and before-after studies generally did not provide sample size justifications, and four of these studies did not account

for confounding variables in patient selection.[15 17 19 20] For the single case-control study, the comparability of cases and controls was scored as 'high risk', due to significant differences in the selection procedure.[26]

## Outcomes and tests used
### Malaria
Three randomised controlled trials (n=20 699) investigated the use of POCT in the context of Malaria.[23–25] All three studies reported the difference in total use of antimalarial drugs between POCT and usual care groups (figure 2). Utilisation of POCT in a pharmacy setting reduced total antimalarial use (risk ratio (RR) 0.58, 95% CI 0.54 to 0.62) over usual care, however pooled estimates exhibited significant statistical heterogeneity ($I^2$=90%). For context, usual care most commonly consisted of pharmacists making decisions to supply antimalarial drugs using their clinical judgement or other parameters such as the patient's temperature, without a rapid diagnostic test.

Two studies reported the difference between appropriately dispensed antimalarial drugs (defined as: antimalaria indicated, antimalarial given) given to patients receiving POCT or usual care.[23 25] These trials found that the risk of receiving inappropriate antimalarial treatment was reduced in the pharmacy POCT group compared with usual care (RR 0.34, 95% CI 0.31 to 0.37, $I^2$=76%).

### International normalised ratio
Five studies (n=1018) investigated the use of POCT in the context of international normalised ratio (INR) testing—four pre/post observational studies,[15–17 27] and one retrospective cohort study (figure 3).[26] Pooled analysis of the pre/post observational studies showed no clear benefit for POCT in pharmacies for INR control, as measured by percentage of time in therapeutic range (TTR) of target INR.[28] Mean difference in percentage TTR between POCT and usual care group was 7.99% (95% CI −0.74% to 16.71%; $I^2$=99%) in favour of pharmacist POCT. The single retrospective cohort study found an increase of 19.90% (95% CI 12.45% to 27.35%) in favour of pharmacist POCT.

### Lipids
Three studies investigated the use of POCT in pharmacies with regards to lipid monitoring. Two studies were pre/post observational studies,[20 21] and one was a randomised trial (online supplementary figure 4).[22]

Total cholesterol (TC) was investigated in all three studies. The RCT showed no significant difference in TC levels over usual care at 6 months (mean difference −7.80, 95% CI −19.65 to 4.05 mg/dL) whereas the pooled analysis of the two observational studies did suggest a significant decrease in TC between baseline and 2-year follow-up (mean difference −29.63 mg/dL, 95% CI −35.29 to −23.98 mg/dL; $I^2$=0%).

At 2 years, meta-analysis showed a significant decrease in low-density lipoprotein (LDL) cholesterol between

**Table 1** Baseline characteristics

| Study author and year | Study design | Country | POC test (condition or level monitored) | Test conducted by | Location of POCT | Total population (n) | Mean (SD) or median age (IQR) (I/C), years | Gender (% M) |
|---|---|---|---|---|---|---|---|---|
| Ansah et al[23] | Cluster RCT | Ghana | Malaria testing (CareStart Malaria HRP2 (Pf), Apacor) | Chemical seller | Private drug retail shops/ chemical shops | 4603 | 15 (6 to 29)/19 (6 to 32) | 51% |
| Mbonye et al[25] | Cluster RCT | Uganda | Malaria testing (First Response Malaria Ag. Combo Rapid Diagnostic Test, Premier Med Corp) | Drug shop vender | 'Drug shops' | 15517 | NA | 48.3% |
| Ikwuobe et al[24] | RCT | Nigeria | Malaria testing (SD Bioline Malaria Antigen Pf, Alere) | Pharmacist | Community pharmacy | 1226 | 30.8 (NA) | 48.3% |
| Al Hamarneh et al[18] | Pre-post observational | Canada | HbA1c (DCA Vantage, Siemens) | Independent prescribing pharmacist | Community pharmacies | 100 | 64 (10.4) | 58% |
| Oyetayo et al[19] | Pre-post observational | USA | HbA1c (device not specified) | Pharmacist | Community pharmacy | 126 | NA | NA |
| Gerrald et al[21] | Pre-post observational | USA | Lipid profile testing (Cholestech LDX Analyzer, Alere) | Pharmacist | Outpatient clinic | 81 | 64.9 (6.9) | 79.1% |
| Peterson et al[22] | RCT | Australia | Total cholesterol (Accutrend GC, Roche Diagnostics) | Pharmacist | Pharmacist visiting at home | 81 | 63.5 (12.1)/65.5 (11.0) | 63% |
| Bluml et al[20] | Pre-post observational | USA | Lipid profile testing (Cholestech LDX Analyzer, Alere) | Pharmacist | Community pharmacy | 397 | 57 (NA) | 48% |
| Deepalakshmi et al[27] | Prospective controlled parallel trial | India | INR (CoaguChek XS Plus, Roche Diagnostics) | Pharmacist | Community pharmacy | 80 | 61.4 (3.1) | 74.4% |
| Harrison et al[15] | Pre-post observational | New Zealand | INR (CoaguChek XS Plus, Roche Diagnostics) | Pharmacist | Community pharmacy | 671 | 72 (13 to 97) | 62.4% |
| Rossiter et al[16] | Pre-post observational | Canada | INR (CoaguCheck XS Machine, Roche Diagnostics) | Pharmacist | Pharmacist-led POC clinic | 119 | 78.8 (NA) | 48.7% |
| Wilson et al[17] | Pre-post observational | USA | INR (Coaguchek-S, Roche Diagnostics) | Pharmacist | Community pharmacy | 19 | 61 (NA) | 68% |

Continued

**Table 1** Continued

| Study author and year | Study design | Country | POC test (condition or level monitored) | Test conducted by | Location of POCT | Total population (n) | Mean (SD) or median age (IQR) (I/C), years | Gender (% M) |
|---|---|---|---|---|---|---|---|---|
| Ernst[26] | Retrospective cohort | USA | INR (CoaguCheck (Boehringer Mannheim) | Pharmacist | Pharmacist led outpatient clinic | 129 | 76.6 (12.8) | 45% |

Studies grouped according to point-of-care test used and chronologically within each test (most recent first).
HbA1c, glycated haemoglobin; I/C, intervention group/control group; INR, international normalised ratio; M, male; NA, not available; Pf, Plasmodium falciparum; POC, point-of-care; POCT, point-of-care test; RCT, randomised controlled trial.

point-of-care and usual care groups (mean difference −28.90 mg/dL, 95% CI −40.74 to −9.65 mg/dL; $I^2$=70%). Furthermore, an increase in high-density lipoprotein (HDL) cholesterol was observed, however this was non-significant (mean difference 3.96 mg/dL, 95% CI −0.80 to +8.72 mg/dL; $I^2$=77%).

Mean TG (triglycerides), LDL and HDL cholesterol were measured in the two observational studies.[20 21] Mean TG concentration was reduced from baseline levels after 2-year follow-up (mean difference −21.68, 95% CI −34.74 to −8.61 mg/dL; $I^2$=0%).

### HbA1c

Two observational studies (n=226) investigated the effect of POCT on HbA1c control among diabetic patients.[18 19] The studies did not find a significant difference between baseline and follow-up HbA1c measurements (−1.02%, 95% CI −2.59% to 0.54%; $I^2$=96%, online supplementary figure 5).

### DISCUSSION
### Summary of findings

We identified 13 studies including over 23 000 patients evaluating the clinical impact of POCT based in pharmacies. The available evidence was generally of poor methodological quality, and only 4 out of 13 studies were randomised controlled trials.

The findings of this review suggest that pharmacy-based POCT may be useful in guiding appropriate anti-malaria prescribing, particularly in low resource settings. Further use of POCT, such as in lipid control, appeared to show some promise although the limited number of studies meant this could not be confirmed and the practical application of these tests in practice were unclear. There was no evidence that the delivery of POCT alone improved INR time-in-therapeutic-range or HbA1c levels in the community pharmacy setting.

### Strengths and limitations

This review provides a timely and comprehensive overview regarding the current evidence related to POCT in pharmacies. The search and review strategy meant that we were unlikely to have missed large numbers of eligible studies. However, studies where the use of POCTs were embedded among other interventions were not included in this review given the difficulty in isolating the effect of the POCT on the outcomes measured. Such studies may have also provide useful information regarding the contribution of POCT to clinical outcomes.

The majority of included studies were observational and were generally of poor methodological quality (online supplementary figures 1-3). Although this limits our understanding of the clinical benefits (or harms) of these POCTs delivered in pharmacies, it highlights a need

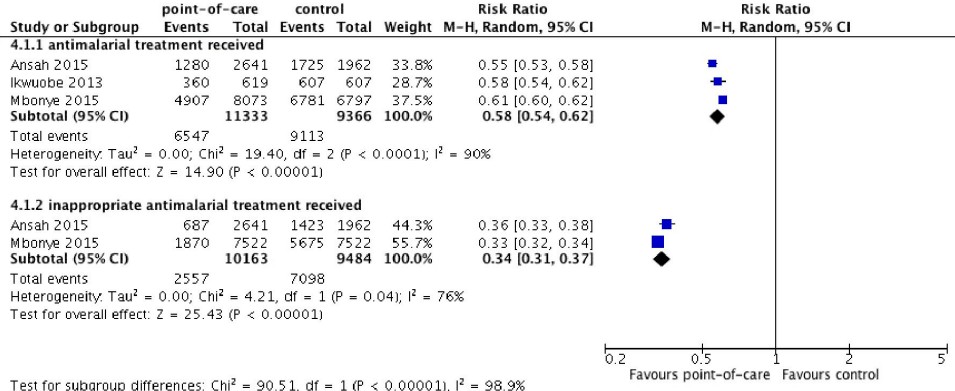

**Figure 2** The effect of pharmacy point-of-care-testing on receiving antimalarial treatment (top) and on the risk of receiving antimalarial treatment when it was not clinically indicated (number of antimalarial medications dispensed).

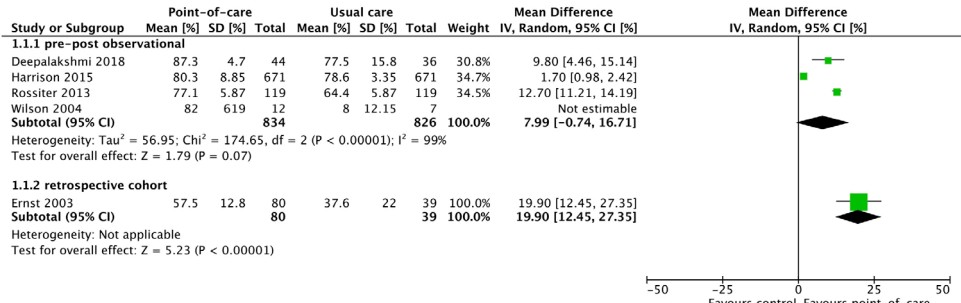

**Figure 3** The effect of pharmacy point-of-care-testing on international normalised ratio (INR) percentage of time in therapeutic range.

for high quality primary studies in this area of clinical practice. Furthermore, the primary literature included in this review were of limited clinical scope, covering only four therapeutic areas (antimalarial drugs, HbA1c, INR and lipid levels). There was no data on areas such as acute infections that commonly present to community pharmacies—something that NHS commissioners have considered introducing into community pharmacies in the UK.[7] There is therefore no strong evidence for the use of POCT for either chronic disease monitoring or acute disease diagnosis in the community pharmacy setting at present. In addition, none of the included studies were conducted in the UK, making the generalisability to UK primary care challenging.

A further limitation of this review concerned the pooling of data from a small number of studies per comparison, leading to high levels of statistical heterogeneity across a majority of comparisons. As a result, the data presented in this systematic review should be interpreted with caution, as the addition of further, larger, studies to this body of evidence are likely to influence these findings.

### Comparison with previous literature
A systematic review published in 2018 by Buss *et al* aimed to summarise the literature related to both analytical quality and effectiveness of POCT in pharmacies.[10] Unlike our review, Buss *et al* included studies where pharmacy POCT performance was compared with corresponding laboratory results. Our review included the two studies contained within Buss *et al* that compared pharmacy POCT with a control.[15 17] In addition to these, we were able to include a further 10 papers.

In 2016, the largest pharmacy chain in the UK conducted a single-arm feasibility study in 35 pharmacies—offering POCT for group A streptococcal pharyngitis.[8] After Centor scoring, patients testing positive for group A streptococci were offered antibiotic treatment at the pharmacy. A total of 149/367 (40.6%) patients received a throat swab, and of these, 36/149 (24.2%) were positive for group A streptococci. Antibiotics were supplied to 9.8% (n=36/367) of patients accessing the service. The study concluded that it was feasible to deliver such a service. The study did not report any clinical outcomes and therefore was not included in this review, although the number of general practitioner

consultations prevented and the reduction in antibiotic use was estimated based on patient self-reporting. Our systematic review has demonstrated that, to date, the impact on both clinical outcomes and total healthcare utilisation are yet to be established with regards to acute bacterial infections from a pharmacy setting.

### Implications for clinical practice
Policymakers have identified community pharmacies as appropriate locations for extended healthcare delivery.[29] This is due in part to the strain on other areas of the health system, the convenience offered to patients who can see a pharmacist without an appointment, and the fact that pharmacists are highly trained in the safe use of medications. However, this systematic review has highlighted that extending the role of pharmacists to delivering POCT may require further assessment before large-scale roll-out. Furthermore, to provide POCT in the future, pharmacists and their staff will require specific training on the tests they provide and in managing the results appropriately. Other considerations such as the practicality and safety associated with handling bodily fluids in pharmacies will also require resolution (eg, a suitable location for patients to provide a urine sample), in addition to having an appropriate medico-legal framework to allow pharmacists to deliver such interventions. Additional considerations include the source of funding for such services (local, national or patient funded). The Strep A feasibility study mentioned above was paid for by patients.[8] How such a model would fit in with the current NHS is another consideration that may require extensive stakeholder deliberation, cost-effectiveness analyses and health inequalities assessment.

Furthermore, the application of the findings from the lipid control studies in this review may be limited, given that in the UK, lipids are most commonly managed on the basis of overall cardiovascular disease (CVD) risk scoring,[30] rather than a stand-alone clinic. Therefore, there is likely to be limited application of POCT for lipids alone, unless it is conducted as part of a CVD risk assessment in the pharmacy.

A policy document presented to the American Pharmacist Association policy committee in 2015 to 2016 outlined the following as potential barriers to the uptake of POCT by pharmacists:[31]

1. Lack of payment mechanisms.
2. Lack of standardised training/education across the profession.
3. Lack of standardised documentation systems and follow-up procedures.
4. Inconsistency in providing POCT services (post-code lottery).
5. Perceived pushback from medical and other related health professionals.

In addition to the operational and practical barriers stated above, this review has highlighted that lack of evidence of effectiveness and healthcare utilisation may also be contributing factors to the lack of commissioning and uptake.

## CONCLUSION AND RECOMMENDATIONS

The few studies available suggest some promise in the use of pharmacy-based POCT for appropriate antimalarial dispensing in low-resource settings, and for the control of blood lipids—however even these results require cautious interpretation given the heterogeneity observed and lack of evidence on clinically relevant outcomes. This systematic review has identified gaps in the literature regarding the evidence for use of POCT in pharmacies, particularly in areas such as the triage and treatment of common acute bacterial or viral respiratory tract infections, where no evidence was found.

Future studies could consider non-inferiority of clinical outcomes versus usual care if the intervention is shown to be safe and cost-effective. Other outcomes such as patient access to care and re-presentation to general practice/out-of-hours care should also be carefully recorded in future studies.

Policy recommendations for the introduction of POCT in pharmacies should be informed by well-conducted randomised controlled trials and economic analyses of each specific condition(s). Until such time as these data become available, caution is required before the widespread roll-out of POCT in pharmacies.

**Acknowledgements** JYJV had full access to all the data in the study and takes responsibility for the integrity of the data and the accuracy of the data analysis.

**Contributors** JYJV and AA conceived the study. AA and JYJV did data extraction. AA and JYJV performed the analyses, which were discussed with AVDB, GH, RJM and JPS. JYJV and AA drafted this report and AVDB, GH, RJM and JPS co-drafted and commented on the final version. All authors had full access to all of the data (including statistical reports and tables) in the study and take responsibility for the integrity of the data and the accuracy of the data analysis. JYJV affirms that the manuscript is an honest, accurate and transparent account of the study being reported; that no important aspects of the study have been omitted and will act as guarantor. All authors have read and approved the final manuscript.

**Funding** JYJV, GH, AVdB are supported through the National Institute for Health Research (NIHR) Community Healthcare MedTech and IVD Co-operative. AA receives funding from the NIHR SPCR. RJM and JS receive support from Oxford NIHR CLAHRC. JS also receives funding from the Wellcome Trust/Royal Society via a Sir Henry Dale Fellowship (ref: 211182/Z/18/Z) and the NIHR School for Primary Care Research. RJM is a NIHR Senior Investigator. The views expressed are those of the authors and not necessarily those of the National Health Service (NHS), the National Institute for Health Research (NIHR), or the Department of Health and Social Care.

**Competing interests** None declared.

**Patient consent for publication** Not required.

**Provenance and peer review** Not commissioned; externally peer reviewed.

**Data availability statement** All data relevant to the study are included in the article or uploaded as supplementary information.

**ORCID iDs**
Ali Albasri http://orcid.org/0000-0001-7805-1965
Gail Hayward http://orcid.org/0000-0003-0852-627X
Richard J McManus http://orcid.org/0000-0003-3638-028X
James Peter Sheppard http://orcid.org/0000-0002-4461-8756
Jan Yvan Jos Verbakel http://orcid.org/0000-0002-7166-7211

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
