## [Reviewer comments · BMJ Open]

ARTICLE DETAILS

TITLE (PROVISIONAL)	Impact of point-of-care tests in community pharmacies: a systematic review and meta-analysis
AUTHORS	Albasri, Ali; Van Den Bruel, Ann; Hayward, Gail; McManus, Richard; Sheppard, James; Verbakel, Jan

VERSION 1 – REVIEW

REVIEWER	Lawrence T Lam University of Technology Sydney, AUSTRALIA Tung Wah College, Hong Kong SAR, CHINA
REVIEW RETURNED	22-Sep-2019

GENERAL COMMENTS	The statistical approach using a random-effect model for the meta-analyses was appropriate. However, there are some minor concerns as follow: 1. Due to the small sample in each of meta-analysis conducted with a high level of heterogeneity in most pooled data, cautions are to apply to the results obtained. This should also be considered as one of the limitation of the study that required some discussions.2. It was acknowledged by the authors that most of the studies included in the meta-analyses were of poorer quality in comparison the the RCTs. Given the methodological shortcomings and the potential of the built-in biases in these studies, one would wonder of the value of subjecting these few studies to a meta-analysis. The authors are suggested to provide a reason for pooling the data of these studies
---

REVIEWER	Frank Moriarty Royal College of Surgeons in Ireland, Ireland
REVIEW RETURNED	30-Nov-2019

GENERAL COMMENTS	This paper reports on a systematic review and meta-analysis on the impact of community pharmacy-delivered POCT on clinical outcomes. This provides very timely evidence on a topic of importance, given the increasing adoption of POCT in this setting. The study appears to have been well conducted and is clearly reported with a thorough discussion of the implications. I have some requests for clarifications and other minor suggested changes that will hopefully further improve an already excellent paper. 1. The title refers to "point-of-care tests by community pharmacists", however the eligibility criteria permit the test being conducted by a member of pharmacy staff, so perhaps "by
--

	community pharmacists" could be amended to "by community pharmacy staff" or "in community pharmacy". 2. The Abstract Conclusion states "Only 4/13 included studies were randomised controlled trials (RCTs) and none were conducted in the UK, limiting our ability to conclusively determine the clinical utility of POCT conducted in pharmacies." This could be rephrases/reordered, as it is not clear why not being conducted in the UK limits the ability to determine clinical utility. (this point is expressed more clearly in the Discussion) 3. The methods provides details of what study designs were included in this review and that systematic reviews were excluded, however the abstract states no study design limits were applied. Please clarify. 4. Page 8 of the Results provides details of charges and recruitment approaches of non-randomised studies, could the same be provided for the included randomised studies? 5. Could the authors clarify the description of the study design for Ernst 2003 (reference 26)? It is described as a retrospective cohort in the results and tables, yet the Methods refers to "case-controlled studies", which I assume refers to this study. From reading the Ernst paper, it does appear to be a retrospective cohort rather than case-control study. 6. The lower bound of the CI for TTR in pre/post studies (page 9, line 43) is missing a minus sign in front of it (0.74%). 7. The discussion states "studies where the use of POCTs were embedded among other interventions were not included in this review" however this is not mentioned in the inclusion/exclusion criteria. 8. There is inconsistency in how online/supplementary figures are referred to. 9. The titles of tables/figures could be amended to be more descriptive.
--	--

VERSION 1 – AUTHOR RESPONSE

Reviewer(s)' Comments to Author:

Reviewer: 1

Reviewer Name: Lawrence T Lam

Institution and Country: University of Technology Sydney, AUSTRALIA; Tung Wah College, Hong Kong SAR, CHINA

Please state any competing interests or state 'None declared': None Declared

Please leave your comments for the authors below

The statistical approach using a random-effect model for the meta-analyses was appropriate.

However, there are some minor concerns as follow:

1. Due to the small sample in each of meta-analysis conducted with a high level of heterogeneity in most pooled data, cautions are to apply to the results obtained. This should also be considered as one of the limitation of the study that required some discussions.

We thank the reviewer for this comment and agree that this is a limitation. We have been cautious with the interpretation of our results and will further highlight this issue in the limitations section of the discussion as the reviewer has suggested.

“A further limitation of this review concerned the pooling of data from a small number of studies for each outcome, leading to high levels of observed statistical heterogeneity. As a result, the data presented in this systematic review should be interpreted with caution, as the addition of further, larger, studies to this body of evidence could influence these findings.”

2. It was acknowledged by the authors that most of the studies included in the meta-analyses were of poorer quality in comparison the the RCTs. Given the methodological short-comings and the potential of the built-in biases in these studies, one would wonder of the value of subjecting these few studies to a meta-analysis. The authors are suggested to provide a reason for pooling the data of these studies

We agree that the lack of eligible studies for each analysis has led to limitations, and we have been cautious not to over-interpret our results. We hope that the graphical representation of the primary studies via forest plots can help the reader to visualise the available data. Furthermore, we have further clarified the decision to pool data, taking into account the expected heterogeneity by means of the random-effects model, in the “Data synthesis”-subheading of the Methods-section:

“Meta-analyses were conducted separately for randomised controlled trials and non-randomised studies whenever three primary studies or more were available per prespecified analysis. Data were analysed using a random-effects model due to expected heterogeneity in study designs and populations”

In answering the comment above, we have reiterated that caution should be taken when interpreting these results, and we conclude the paper with the following paragraph, which we feel reflects the sparse and inconclusive nature of the results:

“Policy recommendations for the introduction of POCT in pharmacies should be informed by well-conducted randomised controlled trials and economic analyses of each specific condition(s). Until such time as these data become available, caution is required before the widespread roll-out of POCT in pharmacies.”

Reviewer: 2

Reviewer Name: Frank Moriarty

Institution and Country: Royal College of Surgeons in Ireland, Ireland

Please state any competing interests or state ‘None declared’: None declared

Please leave your comments for the authors below

This paper reports on a systematic review and meta-analysis on the impact of community pharmacy-delivered POCT on clinical outcomes. This provides very timely evidence on a topic of importance, given the increasing adoption of POCT in this setting. The study appears to have been well conducted and is clearly reported with a thorough discussion of the implications.

I have some requests for clarifications and other minor suggested changes that will hopefully further improve an already excellent paper.

1. The title refers to "point-of-care tests by community pharmacists", however the eligibility criteria

permit the test being conducted by a member of pharmacy staff, so perhaps "by community pharmacists" could be amended to "by community pharmacy staff" or "in community pharmacy".

We thank the reviewer for this comment and agree that this could be clearer. The title now reads "Impact of point-of-care tests in community pharmacies: a systematic review and meta-analysis"

2. The Abstract Conclusion states "Only 4/13 included studies were randomised controlled trials (RCTs) and none were conducted in the UK, limiting our ability to conclusively determine the clinical utility of POCT conducted in pharmacies." This could be rephrased/reordered, as it is not clear why not being conducted in the UK limits the ability to determine clinical utility. (this point is expressed more clearly in the Discussion)

We agree that specific reference to the UK here is not appropriate. We have instead rephrased this in the conclusion of the abstract as below:

"Only 4/13 included studies used a gold-standard randomised controlled trial (RCT) design, limiting our ability to conclusively determine the clinical utility of POCT conducted in pharmacies. Further RCTs are needed, particularly in areas such as upper respiratory tract infections, which have gathered momentum among service commissioners in recent years."

3. The methods provides details of what study designs were included in this review and that systematic reviews were excluded, however the abstract states no study design limits were applied. Please clarify.

We have removed this statement from the abstract now in order to remain consistent with the methods section of this paper. While systematic reviews themselves were excluded, any relevant primary literature they may have contained were screened for possible inclusion.

4. Page 8 of the Results provides details of charges and recruitment approaches of non-randomised studies, could the same be provided for the included randomised studies?

We have now done this and have updated the following paragraph in the study characteristics section of the results, as below:

"None of the RCTs or observational studies charged the patient directly for the POCT. Patients were most commonly recruited into the observational studies via clinician referral or through pharmacy list searches, with only one of the seven observational studies recruiting patients opportunistically. Three of the RCTs recruited patients opportunistically upon presentation to the pharmacy, with the fourth recruiting eligible patients by invitation from a clinical list."

5. Could the authors clarify the description of the study design for Ernst 2003 (reference 26)? It is described as a retrospective cohort in the results and tables, yet the Methods refers to "case-controlled studies", which I assume refers to this study. From reading the Ernst paper, it does appear to be a retrospective cohort rather than case-control study.

This inconsistency has now been corrected. Ernst 2003 is now referred to as a retrospective cohort study in both the results and the methods.

6. The lower bound of the CI for TTR in pre/post studies (page 9, line 43) is missing a minus sign in front of it (0.74%).

We thank the reviewer for picking this up. This now reads -0.74%.

7. The discussion states "studies where the use of POCTs were embedded among other interventions were not included in this review" however this is not mentioned in the inclusion/exclusion criteria.

We have now clarified this in our exclusion criteria and have added the following sentence:

"5. Included a POCT as part of a wider intervention, such that the effect of the POCT alone could not be ascertained."

8. There is inconsistency in how online/supplementary figures are referred to.

We have amended this and we now only refer to them as supplementary figures or tables, as appropriate.

9. The titles of tables/figures could be amended to be more descriptive.

This has now been updated for each figure. For example:

Figure 2 – The effect of pharmacy point-of-care-testing on receiving anti-malarial treatment (top) and on the risk of receiving anti-malarial treatment when it was not clinically indicated (number of anti-malarial medications dispensed).

Figure 3 – The effect of pharmacy point-of-care-testing on International Normalised Ratio (INR) % time in therapeutic range (TTR).

VERSION 2 – REVIEW

REVIEWER	Frank Moriarty Royal College of Surgeons in Ireland, Ireland
REVIEW RETURNED	02-Mar-2020
GENERAL COMMENTS	My thanks to the authors for their responses to my comments and modifications to this paper, this has addressed all of my comments.